# Efficacy and Safety of a Booster Vaccination with Two Inactivated SARS-CoV-2 Vaccines on Symptomatic COVID-19 Infection in Adults: Results of a Double-Blind, Randomized, Placebo-Controlled, Phase 3 Trial in Abu Dhabi

**DOI:** 10.3390/vaccines11020299

**Published:** 2023-01-30

**Authors:** Nawal Al Kaabi, Yunkai Yang, Salah Eldin Hussein, Tian Yang, Jehad Abdalla, Hui Wang, Zhiyong Lou, Agyad Bakkour, Afnan Arafat, Zhiwei Jiang, Ye Tian, Peng Xiao, Walid Zaher, Islam Eltantawy, Chenlong Wang, Guangxue Xu, Yuntao Zhang, Xiaoming Yang

**Affiliations:** 1Sheikh Khalifa Medical City, Abu Dhabi Health Services Company (SEHA), Abu Dhabi 51900, United Arab Emirates; 2College of Medicine and Health Science, Khalifa University, Abu Dhabi 127788, United Arab Emirates; 3China National Biotec Group Company Limited, Beijing 100024, China; 4Beijing Institute of Biological Products Company Limited, Beijing 100176, China; 5MOE Key Laboratory of Protein Science & Collaborative Innovation Center of Biotherapy, School of Medicine, Tsinghua University, Beijing 100084, China; 6Beijing Key-Tech Statistical Consulting Co., Ltd., Beijing 100023, China; 7G42, Abu Dhabi 112778, United Arab Emirates; 8IROS (Insights Research Organization & Solutions), G42 Healthcare, G42, Abu Dhabi 112778, United Arab Emirates; 9National Engineering Technology Research Center for Combined Vaccines, Wuhan Institute of Biological Products Co., Ltd., Wuhan 430207, China

**Keywords:** inactivated vaccines, COVID-19, SARS-CoV-2, booster dose, phase 3 trial

## Abstract

Importance: The protective efficacy of COVID-19 vaccinations has declined over time such that booster doses are required. Objective: To evaluate the efficacy and adverse events of booster doses of two inactivated COVID-19 vaccines. Design: This is a double-blind, randomized, placebo-controlled phase 3 trial aiming to evaluate the protective efficacy, safety, and immunogenicity of inactivated SARS-CoV-2 vaccine (Vero cells) after inoculation with booster doses of inactivated COVID-19 vaccine. Setting: Healthy volunteers were recruited in an earlier phase 3 trial of two doses of inactivated vaccine. The participants in Abu Dhabi maintained the blind state of the trial and received a booster dose of vaccine or placebo at least six months after the primary immunization. Participants: Adults aged 18 and older with no history of SARS-CoV, SARS-CoV-2, or Middle East respiratory syndrome infection (via onsite inquiry) were screened for eligibility. Interventions: A total of 9370 volunteers were screened and randomly allocated, of which 61 voluntarily withdrew from the screening stage without booster inoculation; 9309 received the booster vaccination, with 3083 in the WIV04 group, 3150 in the HB02 group, and 3076 in the alum-only group. Further, 5μg and 4μg of inactivated SARS-CoV-2 virion was adsorbed into aluminum hydroxide in a 0.5 mL aqueous suspension for WIV04 and HB02 vaccines. Main Outcomes and Measures: The primary efficacy outcome was the prevention of PCR-confirmed symptomatic COVID-19 from 14 days after the booster vaccine in the per-protocol population. A safety analysis was performed in the intention-to-treat population. Results: Symptomatic COVID-19 was identified in 36 participants in the WIV04 group (9.9 [95% CI, 7.2–13.8] per 1000 person-years), 28 in the HB02 group (7.6 [95% CI, 5.2–11.0] per 1000 person-years), and 193 in the alum-only group (55.2 [95% CI, 47.9–63.5] per 1000 person-years), resulting in a vaccine efficacy of 82.0% (95% CI, 74.2–87.8%) for WIV04 and 86.3% (95% CI, 79.6–91.1%) for HB02. One severe case of COVID-19 occurred in the alum-only group, and none occurred in the vaccine groups. Adverse reactions within seven days after vaccination occurred in 29.4% to 34.3% of participants in the three groups. Serious adverse events were rare and not related to vaccines (WIV04: 17 [0.5%]; HB02: 11 [0.4%]; alum only: 40 [1.3%]). Conclusions and Relevance: This study evaluated the safety of the booster dose, which was well tolerated by participants. Booster doses given over six months after the completion of primary immunization can help to provide more-effective protection against COVID-19 in healthy people 18 years of age or older. At the same time, the anti-SARS-CoV-2 antibodies produced by the two groups of experimental vaccines exhibited extensive cross-neutralization against representative SARS-CoV-2 variants. Trial Registration: This study is registered on ClinicalTrials.gov (NCT04510207).

## 1. Introduction

The COVID-19 pandemic, caused by severe acute respiratory syndrome coronavirus 2 (SARS-CoV-2), has caused 651 million infections and more than 6.6 million deaths worldwide as of 23 December 2022, according to the WHO COVID-19 Dashboard [1]. People aged 60 years and older and people with pre-existing respiratory or cardiovascular diseases have a high risk of severe disease and death if infected with SARS-CoV-2.

According to the WHO’s draft landscape of COVID-19 candidate vaccines, 147 candidate vaccines are in clinical evaluation, and 195 candidate vaccines are in preclinical evaluation [2]. The candidate vaccines in clinical trials include nucleic acid vaccines, DNA vaccines, replication-defective viral vector vaccines, inactivated pathogen vaccines, protein subunit vaccines, and virus-like particle vaccines [3,4,5,6,7]. Our two inactivated vaccines have demonstrated good protection in completed phase 3 clinical trials [8]. Vaccines with various techniques showed good protective efficacy within a period of time [3,9,10,11]. However, the neutralizing antibody level and the quantity of other immune cells in the vaccines will gradually decline over time [12,13]. According to the World Health Organization (WHO), the COVID-19 vaccination can play a role in population immune protection only after its protective efficiency has reached at least 50% [14]. BNT162b2’s immunoprotective efficacy fell from 88% 1 month after vaccination to 47% at 5–6 months, with the decreasing trend being more pronounced among senior recipients [15]. The vaccine’s effectiveness against symptomatic infections was just 44.1% 20 weeks after two doses of the ChAd0x1-S vaccine, according to a study by the UK health security agency [16]. Therefore, it is suggested that booster shots are required. Previous studies had validated inactivated SARS-CoV-2 vaccines (Vero cells) for efficacy and safety [11,17]. Here we report the protective efficacy, safety, and immunogenicity of inactivated SARS-CoV-2 vaccines (Vero cells) after inoculation with booster doses in a double-blind, randomized, placebo-controlled phase 3 trial in Abu Dhabi.

## 2. Methods

### 2.1. Study Design and Participants

This double-blind, randomized, placebo-controlled, phase 3 trial was designed by the Wuhan Institute of Biological Products Co., Ltd., Wuhan, China and the Beijing Institute of Biological Products Co., Ltd., Beijing, China both of which belong to the China National Biotec Group Company Limited, Beijing, China. The ongoing trial is being performed and data collected by the investigators at the Sheikh Khalifa Medical City in Abu Dhabi. An independent data and safety monitoring board is monitoring safety data and evaluating risks among the participants. The study protocol was approved by the UAE Ministry of Health and Prevention. Written informed consent was obtained from all participants before enrollment.

Adults aged 18 and older with no history of SARS-CoV, SARS-CoV-2, or Middle East respiratory syndrome infection (via onsite inquiry) were screened for eligibility. Exclusion criteria included (but were not limited to) pregnancy or breastfeeding; known allergy to components of the study vaccine or placebo; recent (within the past 6 months) or planned use of immunosuppressive therapy or use of immunoglobulins or any blood products within the past 3 months; history of a bleeding disorder; and any confirmed or suspected autoimmune or immunodeficiency disease. Detailed inclusion and exclusion criteria are shown in the eMethods in the Appendix A.

### 2.2. Randomization and Masking

Participants were randomly assigned to 1 of the 2 vaccine groups or a control group receiving aluminum hydroxide (alum) adjuvant only (in a 1:1:1 ratio), according to unique serial numbers generated by an independent statistician. The concealed random grouping allocation and blind codes were blinded to the investigators, participants, and statisticians. The masking was removed in the event of a medical emergency requiring acute intervention, upon the responsible investigator’s approval and the data and safety monitoring board’s knowledge. The participants received 2 intramuscular vaccinations 21 days apart and another booster shot after at least 6 months.

### 2.3. Trial Vaccine Information

Development of the 2 inactivated vaccines has been previously described. In brief, 2 SARS-CoV-2 strains (WIV04 and HB02) were isolated from 2 patients in Jinyintan Hospital, Wuhan, China, and separately used to develop the 2 vaccines (referred to as WIV04 and HB02 vaccines, hereafter). The virus strains were cultivated in qualified Vero cell lines for proliferation, followed by purification and beta propiolactone inactivation. The final product contained 5 μg virion adsorbed with 0.5 mg of aluminum hydroxide in WIV04 vaccine and 4 μg virion adsorbed with 0.225 mg of aluminum hydroxide in HB02 vaccine. Each was packed into prefilled syringes in 0.5-mL sterile phosphate-buffered saline without preservative. The same dose of alum adjuvant (0.5 mg) was selected as placebo control and packed into prefilled syringes in 0.5 mL of sterile phosphate-buffered saline without preservative. The vaccines and controls were approved by the National Institutes for Food and Drug Control of China and were supplied in coded, identical-appearing, single-dose syringes.

### 2.4. Assessments

Participants were educated about COVID-19 and its symptoms, and their progress was tracked through active investigator follow-up (weekly phone calls and a face-to-face examination at the time of vaccination), active participant and sentinel hospital reports, and passive linkage to the local medical system. We defined a COVID-19 suspected case as either epidemic history with two or more A symptoms for at least 2 days, specifically fever (axillary temperature more than 37.5 °C), chills, sore throat, stuffy nose, myalgia, fatigue, headache, nausea or vomiting, or diarrhea; with one or more B symptoms, specifically cough (presence at least 2 days), shortness of breath, new olfactory or taste disorder (presence at least 2 days); or with radiographic evidence of COVID-19–like pneumonia. Finally, with a positive PCR testing result, the participant shall be considered as a confirmed COVID-19 case. According to the National Health Commission of China’s Diagnosis and Treatment Scheme for COVID-19, cases were diagnosed and severity status was classified as mild, moderate, severe, or critical, with modifications by referring to the World Health Organization (WHO) and the US Centers for Disease Control and Prevention criteria (eMethods).

The case diagnosis and severity were independently determined by an end point assessment and adjudication committee (EAC) comprising specialists in related fields who were unaware of their group assignment at the time of adjudication.

### 2.5. Efficacy End Points

The primary efficacy end point was a laboratory-confirmed symptomatic COVID-19 case that occurred at least 14 days after the booster vaccine. The secondary efficacy end point was the occurrence of severe COVID-19 and/or death at least 14 days after booster vaccination. Two exploratory post hoc efficacy analyses included the protective effect of neutralizing antibodies against COVID-19 and the occurrence of ADE/VED after booster vaccine. The definition and description of the analysis set are in the Appendix A.

### 2.6. Safety End Points

Within 7 days of each vaccination, participants were asked to note on diary cards any site-specific adverse reactions (e.g., pain, redness, swelling) and any systemic adverse reactions (e.g., fever, headache, fatigue) to vaccination. During the follow-up, any new unanticipated symptoms and indicators were also noted. The investigators decided on the grading criteria for adverse events and the causality to vaccination before unblinding, and information can be found in the protocol (Appendix A) and earlier publications.

### 2.7. Exploratory Immunogenicity End Point

Blood samples were collected 14 days, 28 days, and 6 months after booster vaccination. Serum samples from the first 900 participants were selected for the immunogenic subgroup tested for neutralization ability using live SARS-CoV-2 virus (strain 19nCoVCDC-Tan-Strain04 [QD01]) at 50% cell culture infectious dose. Details of the immunogenicity assay have been previously described and are provided in eMethods. A positive antibody response (seroconversion) was defined as at least 1:4 if the baseline titer was less than 1:4 or at least a 4-fold rise from baseline if the baseline titer was at least 1:4.

### 2.8. Neutralizing Activity of Serum Samples

On the premise of maintaining a blinded state, from the serum samples of the phase 3 clinical trial submitted for inspection, the first 150 serum samples (numbers B00001-2 to B00164-2) of the immunogenic subgroups were sequentially selected according to the serum number for the cross-neutralization analysis of ancestral strains (10 in total) in different regions and another 150 samples (No. B00165-2 to B00338-2) for cross-neutralization analysis of variant strains (Beta and Delta). Neutralizing antibody titer <1:4 is negative, and ≥1:4 is positive.

### 2.9. Statistical Analysis

The vaccine efficacy in confirmed COVID-19 cases of the 2 vaccine groups compared with the alum-only group and their 95% confidence interval (CI) were calculated from the exact Poisson regression model. The model included the number of confirmed COVID-19 cases as the dependent variable, the treatment group as the independent variable, and person-years as the offset. The primary efficacy analysis was performed from the modified full analysis population, which included participants who received the booster dose, had at least 1 efficacy follow-up visit after day 14 following the booster dose. The participants who were diagnosed with COVID-19, whether PCR positive or vaccinated with other COVID-19 vaccines within 14 days after booster dose, were excluded from the primary analysis population. The subgroup analyses were conducted in each stratum of age (18~59 yrs vs. ≥60 yrs), where the time interval between the 2nd dose and the booster dose (<6 months, ≥6 months, 6~<9 months, and ≥9 months) were used in post hoc analyses.

The incidence rates of adverse events and reactions in each group are described in the total cohort of participants who received the booster dose. All analyses were conducted by using SAS software, version 9.4 (SAS Institute Inc., 100 Sas Campus Dr, Cary, NC 27513).

## 3. Results

### 3.1. Study Participants

Trials are intended to be conducted on an informed and voluntary basis; it is challenging to maintain placebo groups given the widespread usage of vaccines. Therefore, the experiment was carried out only in Abu Dhabi, UAE, and 9370 volunteers who had participated in the previous phase 3 clinical trial volunteered to participate in the experiment and kept the original grouping on a blinded basis: 3097 in the WIV04 group (WIV04-CoV vaccine group), 3172 in the HB02 group (BBIBP-CoV vaccine group), and 3101 in the alum-only group (Figure 1). Among all participants, 9309 received the booster vaccination (3083 in the WIV04 group, 3150 in the HB02 group, and 3076 in the alum-only group). Some participants are still being followed up with, among whom 9071 (3006 in the WIV04 group, 3055 in the HB02 group, and 3010 in the alum-only group) had negative PCR test results before the booster dose and were included in the modified full analysis set (mFAS) for the primary efficacy analysis.

Baseline characteristics in the modified full analysis population are shown in Table 1. The mean (SD) age of participants was 28.8 (9.8) years, and 7725 (85.2%) were men. During the trial period, participants had been residing in the UAE, and their original nationalities, according to passport information, comprised the UAE (35.2%), India (15.0%), Pakistan (7.6%), Bangladesh (7.3%), the Philippines (6.5%), Egypt (5.3%), Nepal (3.1%), China (2.8%), Syria (2.5%), and Yemen (1.8%). Of the study samples, 85.2% were from men and 64.8% were from expats. The situation is influenced by the UAE’s demographics, which are dominated by men (72.0%) and expats (88.5%). The interval between a booster and basic immunization was longer than anticipated because of the experiment’s size and logistical challenges.

### 3.2. Primary Efficacy End Point

The cutoff date for analyzing the booster’s protective efficacy was 17 October 2021. In this study, 347 symptomatic COVID-19 cases were gathered 14 days following booster vaccination, of which 262 were confirmed by EAC. The EAC recognized 257 valid cases after an impartial review, including 18 severe cases, 36 moderate cases, and 205 mild cases. There were 36 cases in the WIV04 group, 28 cases in the HB02 group, and 193 cases in the placebo group. Vaccine efficacy analysis was based mainly on the mFAS analysis set.

The incidence rate (per 1000 person-years) was 9.9 (95% CI, 7.2–13.8) in the WIV04 group, 7.6 (95% CI, 5.2–11.0) in the HB02 group, and 55.2 (95% CI, 47.9–63.5) in the alum-only group. This resulted in a vaccine efficacy, compared with the alum-only group, of 82.0% (95% CI 74.2–87.8%) for the WIV04 group and 86.3% (95% CI, 79.6–91.1%) for the HB02 group (Table 2 and Figure 2). Vaccine efficacy was similar in the sensitivity analyses among various study populations (Appendix A in Appendix A).

### 3.3. Secondary Efficacy End Point

Among the valid COVID-19 cases, 16 severe cases were in the alum-only group, none was in the WIV04 group, and 1 severe case was in the HB02 group, resulting in a vaccine efficacy of 100% (95% CI 74.9–100.0%) and 94.1% (95% CI 79.6–91.1%) against severe COVID-19 in the vaccine groups, respectively (Table 2). However, these results should be interpreted with caution given the small number of severe cases.

### 3.4. Post Hoc Analyses

A total of 257 symptomatic cases of COVID-19 were monitored in the study population from 14 days following the booster dose given at interval of ≥6 months from the primary immunization: 36 in the WIV04 group, 28 in the HB02 group, and 193 in the alum-only group, resulting in a vaccine efficacy of 82.0% (95% CI, 74.2–87.8%) for the WIV04 vaccine and 86.3% (95% CI 79.6–91.1%) for the HB02 vaccine (Table 2). The incidence rates were lower in the vaccine groups compared with the alum-only group. In post hoc subgroup analyses, similar vaccine efficacy was observed at different study sites and for men and women (Appendix A). In total, 338 participants were 60 years of age and older, with 115 in the WIV04 group, 99 in the HB02 group, and 124 in the alum-only group. One COVID-19 case occurred in the WIV04 group, one case in the HB02 group, and three cases in the alum-only group, resulting in a vaccine efficacy of 61.5% (95% CI −379.5–99.3%) and 60.0% (95% CI −399.1–99.2%) (Appendix A). However, additional data were needed in order to obtain more-reliable results, as there were fewer elderly participants.

### 3.5. Exploratory Immunogenicity End Points

The geometric mean titers (GMTs) of the neutralizing antibody with in an iFAS analysis set before the booster dose were 47.6 in the WIV04 group, 97.2 in the HB02 group, and 4.5 in the alum-only group, while neutralizing antibody GMTs on 14 days after the booster dose were 192.4 (95% CI, 166.8–221.8) in the WIV04 group, 492.0 (95% CI, 444.4–544.7) in the HB02 group, and 4.4 (95% CI, 3.8–5.1) in the alum-only group. On 28 days after the booster dose, neutralizing antibody GMTs were 280.8 (95% CI, 243.1–324.5) in the WIV04 group, 813.9 (95% CI, 726.5–911.8) in the HB02 group, and 5.1 (95% CI, 4.4–45.9) in the alum-only group (Figure 3). Neutralizing antibody levels were significantly higher in the HB02 vaccine group than in the WIV04 group, regardless of gender or time point differences (Appendix A).

### 3.6. Neutralizing Activity of Serum Samples

The seropositive rates of neutralizing antibodies against 10 ancestral strains of serum samples from the WIV04 and HB02 groups were both 100%, while the seropositive rates of serum samples from the placebo control group ranged from 7.6 to 15.1% (Appendix A).

The 49 serum samples in the WIV04 group all produced high neutralizing antibodies against the three strains, among which the neutralizing antibody GMT of the ancestral strain (QD strain) was 89.3 (95% CI, 72.1–110.6), and the neutralizing antibody GMTs of the mutant Beta and Delta strains were 46.8 (95% CI, 38.5–56.9) and 63.1 (95% CI, 52.6–75.6), respectively, and the neutralizing antibody levels against the Beta and Delta strains reached 52.4% and 70.7% of the antibody levels of the ancestral strain, respectively. In the HB02 group (54 serum samples), the neutralizing antibody GMT of the ancestral strain was 150.2 (95% CI, 126.5–178.4), and the neutralizing antibody GMTs of the mutant Beta and Delta strains were, respectively, 78.1 (95% CI, 66.8–91.4) and 106.6 (95% CI, 88.7–128.2). The neutralizing antibody levels to the Beta and Delta strains reached 52.0% and 71.0% of the ancestral strains, respectively (Table 3), indicating that the WIV04 and HB02 immune sera had a good cross-neutralizing ability against different mutant strains.

### 3.7. Adverse Reactions and Events

Systemic and injection-site reaction symptoms are shown in Table 4. Within 28 days after booster dose vaccination, the total adverse reactions were reported by 1113 participants (1113/3083) in the WIV04 group, 992 (992/3150) in the HB02 group, and 1132 (1132/3076) in the alum-only group. The most common adverse reaction in the WIV04 group, HB02 group, and alum-only group was pain at the injection site (23.3%, 17.2%, and 21.3%), followed by headache (7.2%, 8.4%, and 7.3%). Most adverse reactions were mild in severity (grade 1 or 2) and were transient and self-limiting, without need for special treatment.

Unsolicited adverse events (related or unrelated to vaccination) are shown in Table 4 (more-detailed information can be found in the Appendix A). Adverse events were reported by 10.5% of the participants in the WIV04 group, 10.8% in the HB02 group, and 12.9% in the alum-only group.

A total of 68 serious adverse events occurred during follow-up in the three groups: 0.6% in the WIV04 group, 0.4% in the HB02 group, and 1.3% in the alum-only group (Appendix A). No adverse event was considered to be possibly related to the vaccination.

## 4. Discussion

COVID-19 vaccinations are critical for preventing infection, regulating transmission, preventing serious infections, and lowering mortality [3]. Despite the fact that the vaccine’s effectiveness as a preventive measure diminishes over time, a booster dose offers superior protection. In this phase 3 randomized trial in adults, booster doses of inactivated COVID-19 vaccines showed efficacy levels of 82.0% and 86.3% against symptomatic COVID-19 cases. The two vaccines had rare serious adverse events at a frequency similar to the alum-only control, and the majority were not related to the vaccinations.

Clinical results of booster vaccination with other vaccines have recently been published. A homologous booster of BNT162b2 increased neutralizing antibody titers against the wild-type virus and the Delta variant to more than five times as high as after the second dose [18]. In addition, 101 solid organ transplant recipients who received three doses of BNT162b2 (Pfizer-BioNTech) were examined, and the examination discovered that detectable antispike antibody levels rose from 40% to 68% [19]. In another study with 96 heart transplant recipients utilizing neutralizing antibodies against wild-type viruses, the authors noticed an increase in neutralizing titer of more than 9.0-fold [20]. Additionally, a third dose booster of a homologous/heterologous vaccination provided a significant increase in antibody responses after two doses of CoronaVac [21]. Another work indicated that a mRNA-1273 booster injection increased neutralizing antibody titers against a wild-type virus and the Delta variant 3.8-fold and 2.1-fold, respectively, compared to the primary series [22]. Our study showed NAbs levels in the WIV04 and HB02 groups decreased 2.5 times and 2.6 times before the booster dose, respectively, compared to 28 days after the second dose, then climbed 5.9 times and 8.4 times, respectively, 28 days after the booster dose. However, binding and neutralizing antibodies may not be sufficient indicators of protection against these more-serious outcomes. Additional clinical research is required to assess the vaccine effectiveness. Another study found a vaccine effectiveness rate of 89% with an additional dose of BNT162b2 vaccine [23], but there was no placebo group.

Our research is a long-term, continuous, large-scale effort. The blind state maintained through the entire phase 3 clinical trial of two doses of inactivated vaccine and the phase 3 clinical trial of the booster vaccination. The vaccine effectiveness at all stages was counted (Appendix A). The booster dose’s vaccine effectiveness returned to above 80%: 1.6 times and 1.5 times greater than 7 months after the primary immunization, respectively. The differences of the protection effect and the GMT level of the two vaccines may be due to the different inactivation process. All adverse events were common mild adverse events, and none of the adverse events was considered to be possibly related to the vaccination, which illustrates the safety of the vaccines.

There is growing interest in investigating whether vaccines can protect against emerging SARS-CoV-2 variants, especially for the variants of concern [24,25,26,27]. A narrative review [28] yields a comparative analysis of the efficacy and effectiveness of COVID-19 vaccines against the variants of concern, which showed that most COVID-19 vaccines had high efficacy against the ancestral strain and against the Alpha, Beta, Gamma, and Delta variants, but the quality of evidence greatly varies depending on the vaccine of interest. A study found no protection offered from the ChAdOx1 nCoV-19 vaccine against the B.1.351 variant [29]. Surprisingly, in the majority of vaccines, whether the vaccine neutralized the Omicron variant was undetectable [30]. The blind sampling of participants in this phase 3 clinical trial showed that through the cross-neutralization experiment, both WIV04 and HB02 vaccines have good cross-neutralizing responses to the SARS-CoV-2 ancestral strains and the Beta and Delta variants. However, more research is still needed to investigate the neutralizing ability and efficacy of inactivated vaccines against emerging SARS-CoV-2 variants.

This study also has several limitations. First, the study did not include pregnant women or those younger than 18 years; thus, the efficacy and the safety of the inactivated vaccines in these groups remain unknown. Results from phase 1/2/3 trials among these groups have not been reported, but an evaluation of the safety and immunogenicity of the two vaccines among children and adolescents is ongoing. Second, the trial was conducted mainly in generally healthy young men in the Middle East, and there was insufficient ability to test the efficacy among those with chronic diseases, women, older adults, those in other geographic populations, and those with previous SARS-CoV-2 infections; this includes people who are most vulnerable to severe COVID-19 cases and mortality. Third, the study could not address the question whether the inactivated vaccines prevent asymptomatic infection, which requires formal study-wide surveillance via virologic and serologic tests.

## 5. Conclusions

This study evaluated the safety of the booster dose, which was well tolerated by participants. Booster doses given over 6 months after the completion of primary immunization can help to provide more-effective protection against COVID-19 in healthy people 18 years of age or older. At the same time, the anti-SARS-CoV-2 antibodies produced by the two groups of experimental vaccines exhibited extensive cross-neutralization against representative SARS-CoV-2 variants.

## Figures and Tables

**Figure 1 vaccines-11-00299-f001:**
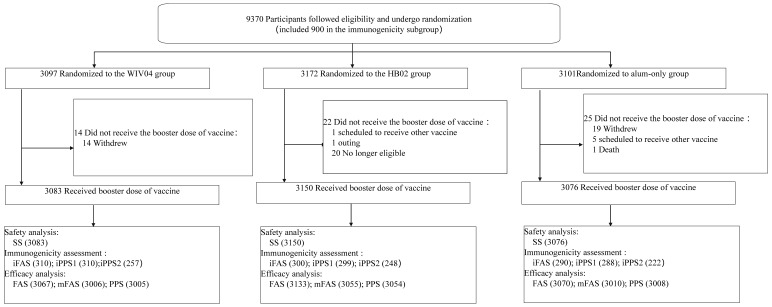
Flow of Participants. See eMethod in Appendix A for definitions of each analysis population.

**Figure 2 vaccines-11-00299-f002:**
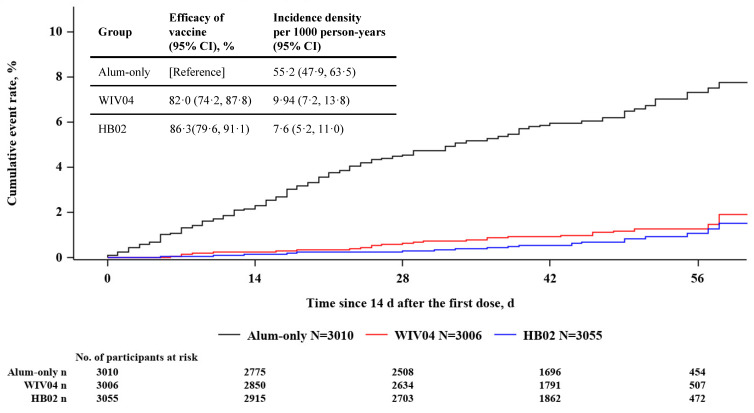
**Efficacy of 2 Inactivated Vaccines Against Symptomatic COVID-19**. Cumulative event rates of confirmed symptomatic COVID-19 cases 14 days following a booster dose among participants who contributed at least 1 efficacy follow-up visit, and had negative polymerase chain reaction test results at enrollment (modified full analysis set).

**Figure 3 vaccines-11-00299-f003:**
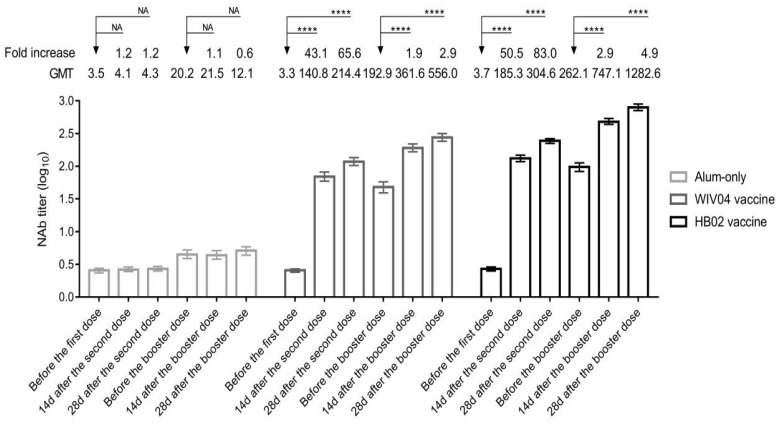
**Neutralizing Antibodies to Live SARS-CoV-2 Before and After Booster Dose**. Seroconversion was defined as postinjection titer of at least 16 if the baseline titer was below 4 or at least a 4-fold increase in postinjection titer from baseline if the baseline titer was at least 4. The 95% CIs were calculated using the Clopper-Pearson method. **** *p* < 0.0001.

**Table 1 vaccines-11-00299-t001:** Baseline Characteristics of Participants.

	WIV04 Vaccine Group (n = 3067)	HB02 Vaccine Group (n = 3133)	Alum-Only Group (n = 3070)	Total (n = 9270)
Age, mean (SD), y	38.9 (9.8)	38.8 (9.8)	38.7 (9.9)	38.8 (9.8)
Age group				
<60 y	2946 (96.1)	3029 (96.7)	2946 (96.0)	2946 (96.0)
≥60 y	121 (4.0)	104 (3.3)	124 (4.0)	124 (4.0)
Sex				
Male	2594 (84.6)	2669 (85.2)	2639 (86.0)	2639 (86.0)
Female	473 (15.4)	464 (14.8)	431 (14.0)	431 (14.0)
Study sites				
Abu Dhabi	3067	3133	3070	3070
National origin				
United Arab Emirates	1104 (36.0)	1087 (34.7)	1033 (33.7)	1033 (33.7)
India	449 (14.6)	474 (15.1)	472 (15.4)	472 (15.4)
Bangladesh	242 (7.9)	224 (7.2)	230 (7.5)	230 (7.5)
Pakistan	228 (7.4)	250 (8.0)	236 (7.7)	236 (7.7)
Philippines	180 (5.9)	218 (7.0)	206 (6.7)	206 (6.7)
Egypt	169 (5.5)	167 (5.3)	161 (5.2)	161 (5.2)
Nepal	89 (2.9)	84 (2.7)	111 (3.6)	111 (3.6)
China	85 (2.8)	93 (3.0)	84 (2.7)	84 (2.7)
Syria	74 (2.4)	67 (2.1)	89 (2.9)	89 (2.9)
Jordan	68 (2.2)	63 (2.0)	80 (2.6)	80 (2.6)
Others	379 (12.4)	406 (13.0)	368 (12.0)	368 (12.0)
Exposure time after booster vaccination, mean (SD), d	57.4 (15.5)	57.6 (15.2)	56.0 (16.4)	56.0 (16.4)
Interval between booster vaccination and basic immunization mean (SD) (month) ^a^	9.6 (0.7)	9.6 (0.7)	9.6 (0.7)	9.6 (0.7)
Interval between booster vaccination and basic immunization				
<6 months	9 (0.3)	12 (0.4)	11 (0.4)	11 (0.4)
6–9 months	144 (4.7)	127 (4.1)	134 (4.4)	134 (4.4)
≥9 months	2914 (95.0)	2994 (95.6)	2925 (95.3)	2925 (95.3)

Abbreviation: alum, aluminum hydroxide. WIV04 and HB02 groups represent the vaccine groups which were developed based on two SARS-CoV-2 strains, WIV04 and HB02 strains. The analyses were conducted in the full analysis set included those who received the booster dose, contributed at least 1 efficacy follow-up visit, and had negative polymerase chain reaction test results at enrollment. a: Interval between booster vaccination and basic immunization (months) = (booster vaccination date − second dose vaccination date in the basic immunization)/30.4375.

**Table 2 vaccines-11-00299-t002:** Efficacy of 2 Inactivated Vaccines Against Symptomatic COVID-19.

Incident COVID-19 Cases and Vaccine Efficacy 14 Days after Booster Doses of Immunization ^a^			
Outcome	WIV04 Vaccine Group	HB02 Vaccine Group	Alum-Only Group
**Primary analysis: incident symptomatic cases**			
No. of participants	3006	3055	3010
No. of incident cases	36	28	193
Person-years	362.3	370.4	349.8
Incidence density per 1000 person-years (95% CI)	9.94 (7.2, 13.8)	7.6 (5.2, 11.0)	55.2 (47.9, 63.5)
Vaccine efficacy (95% CI), %	82.0 (74.2, 87.8)	86.3 (79.6, 91.1)	[Reference]
**Secondary analysis: incident severe cases**			
No. of participants	3006	3055	3010
No. of incident cases	0	1	16
Person-years	362.4	370.5	350.4
Incidence density per 1000 person-years (95% CI)	0 (0.0, 1.0)	0.3 (0.0, 1.50)	4.57 (2.6, 7.4)
Vaccine efficacy (95% CI), %	100.0 (74.9, 100.0)	94.09 (62.0, 99.9)	[Reference]
**Post hoc analysis: incident symptomatic and asymptomatic cases**			
**Analysis of protective efficacy in population with interval between booster vaccination and basic immunization ≥ 6 months**			
No. of participants	2997 ^b^	3043 ^b^	2999 ^b^
No. of incident cases	36	28	193
Person-years	361.4	369.2	348.6
Incidence density per 1000 person-years (95% CI)	10.0 (7.2, 13.1)	7.6 (5.2, 11.0)	55.4 (48.1, 63.8)
Vaccine efficacy (95% CI), %	82.0 (74.2, 87.8)	86.30 (79.6, 91.1)	[Reference]

Abbreviation: alum, aluminum hydroxide; a. The analyses were conducted in the modified full analysis set. A Poisson regression model with log-link function was used, with the number of incident cases as the dependent variable, treatment group as the independent variable, and person-years as the offset. Incidence density with its 95% CI was estimated using the least-square method. If the number of cases in any of the groups was less than 5, the exact method was used to estimate the incidence rate, vaccine efficacy, and 95% CI using StatXact software. b. 32 participants’ interval between booster vaccination and basic immunization was less than 6 months.

**Table 3 vaccines-11-00299-t003:** Neutralizing antibody to various strains of SARS-CoV-2.

Virus Strain	GMT	WIV04 Group (n = 49)	HB02 Group (n = 54)	Alum Only Group (n = 47)
QD01	GMT (95% CI)	89.3 (72.1, 110.6)	150.2 (126.5, 178.4)	2.1 (2.0, 2.3)
	neutralizing antibody levels compared with prototype strains	/	/	/
Beta	GMT (95% CI)	46.8 (38.5, 56.9)	78.1 (66.8, 91.4)	2.1 (2.0, 2.1)
	neutralizing antibody levels compared with prototype strains	52.4%	52.0%	/
Delta	GMT (95% CI)	63.1 (52.6, 75.6)	106.6 (88.7, 128.2)	2.1 (2.0, 2.3)
	neutralizing antibody levels compared with prototype strains	70.7%	71.0%	/

Abbreviation: alum, aluminum hydroxide; WIV04 and HB02 groups represent the vaccine groups which were developed based on two SARS-CoV-2 strains, WIV04 and HB02 strains. WIV04: hCoV-19/Wuhan/WIV04/2019; HB02: 19nCoV-CDC-Tan-HB02. QD01: 19nCoV-CDC-Tan-Strain04 (QD01); Beta: hCoV-19/Guangzhou/IVDC-GDPCC-nCoV84/2021; Delta: NCoV210077;

**Table 4 vaccines-11-00299-t004:** Common Adverse Reactions and Grades in the Safety Analysis Set.

Adverse Event	WIV04 (n = 3083)	HB02 (n = 3150)	Alum Only (n = 3076)
Total adverse events	1113 (36.1)	992 (31.5)	1132 (36.8)
Solicited adverse events	1004 (32.6)	871 (27.7)	950 (30.9)
Local events	736 (23.9)	556 (17.7)	672 (21.9)
Pain	718 (23.4)	543 (17.2)	656 (21.3)
Systemic events	460 (14.9)	513 (16.3)	444 (14.4)
Fever	67 (2.2)	64 (2.0)	53 (1.7)
Diarrhea	34 (1.1)	34 (1.1)	27 (0.9)
Myalgia (Non-inoculated site)	118 (3.8)	123 (3.9)	112 (3.6)
Headache	221 (7.2)	264 (8.4)	223 (7.3)
Coughing	70 (2.3)	75 (2.4)	67 (2.2)
Fatigue	143 (4.6)	178 (5.7)	137 (4.5)
Unsolicited adverse reactions	323 (10.5)	340 (10.8)	398 (12.9)

Data are shown as No. of participants with event (%). The safety analysis population includes all subjects who received the booster dose of vaccine. Participants withmore than 1 adverse reaction in a specific reaction category were only counted once; for example, if they had the same symptom (e.g., injection-site pain) after each dose or if they had more than 1 symptom in the reaction class (total, systemic, and local), they were only counted once in that adverse reaction class. Participants with both lower- and higher-grade adverse events were counted once in the higher-grade total adverse events. Grading scales for systemic andlocal adverse events are detailed in the eMethod in Appendix A.

## Data Availability

We support data sharing of individual participant data. After de-identification, the individual participant data (i.e., text, tables, graphs, and supplement) underlying the results reported in this article will be shared. Individual participant data will be available beginning 3 months and ending 1 year after publication. Supporting clinical documentation, including study protocol, statistical analysis plan, and informed consent, will be available for at least 1 year immediately after publication. Researchers providing scientifically sound advice will be given access to individual participant data. Please send proposals to yangxiaoming@sinopharm.com. These proposals will be reviewed and approved by funders, investigators, and collaborators according to scientific merit. To gain access, data requesters will need to sign a data access agreement.

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
