# Peer review of "Efficacy and Safety of a Booster Vaccination with Two Inactivated SARS-CoV-2 Vaccines on Symptomatic COVID-19 Infection in Adults: Results of a Double-Blind, Randomized, Placebo-Controlled, Phase 3 Trial in Abu Dhabi"

_vaccines, 2023, doi:10.3390/vaccines11020299_

Round 1

Reviewer 1 Report

The authors have presented the study of "Efficacy and safety of a booster vaccination with 2 inactivated SARS-CoV-2 vaccines on symptomatic COVID-19 infection in adults :results of a double-blind,randomised, placebo-controlled, phase 3 trial in Abu Dhabi". The manuscript is well written with minor spell checks. However, few concerns are:

1. Line61 data needs to be updated.

2. The strains used in these vaccines must be mentioned.

3. In line 123 it should be beta propiolactone in place of β-propanolide.

4. Only one alum control was used in the study i.e. for WIV-4 and not for HB02 as the concentration of alum is different in two vaccines.

5. If possible neutralization with omicron strain may be done. As this is more prevalent now.

6. In discussion authors should include more discussion with inactivated vaccines. 

7. Participants of age 60 and above are less so data may be misleading

Author Response

The authors have presented the study of "Efficacy and safety of a booster vaccination with 2 inactivated SARS-CoV-2 vaccines on symptomatic COVID-19 infection in adults :results of a double-blind,randomised, placebo-controlled, phase 3 trial in Abu Dhabi". The manuscript is well written with minor spell checks. However, few concerns are:

Our response: We thank the reviewer for the summary of our study, as well as the positive comments.

  1. Line61 data needs to be updated.

Our response: We appreciate the reviewer's suggestion and have updated according to the latest data at the time of modification.

  1. The strains used in these vaccines must be mentioned.

Our response: We appreciate the reviewer's recommendation and the strains have been indicated in the legends of Table 3 and eTable 4, where we also displayed other viral strains implicated

  1. In line 123 it should be beta propiolactone in place of β-propanolide.

Our response: We accepted the reviewer's comment and made corrections

  1. Only one alum control was used in the study i.e. for WIV-4 and not for HB02 as the concentration of alum is different in two vaccines.

Our response: We thank the reviewer for his/her question. Different adjuvant producers were used because the two vaccines were created independently by two different businesses. As a result, adjuvant concentration is not the same. Since Wuhan Institute of Biological Products supplied the adjuvant for the control group in the trial, its product WIV04 served as the standard for adjuvant concentration.

  1. If possible neutralization with omicron strain may be done. As this is more prevalent now.

Our response: We strongly agree with the reviewer that neutralization with omicron strain is very meaningful, but unfortunately the experiment could not be carried out due to the amount of serum, which is also our great regret.

  1. In discussion authors should include more discussion with inactivated vaccines. 

Our response: We agree with the reviewers and include a summary highlighting the effectiveness of inactivated vaccines as booster doses. And we believe it's crucial to compare the booster doses' protective effects to those of other vaccines.

  1. Participants of age 60 and above are less so data may be misleading

Our response: We agree with the reviewer that the participants of age 60 and above are less, so we also advise caution in drawing conclusions. However, there are few studies on such people, so we present the results for reference.

Reviewer 2 Report

Kaabi et al of the manuscript titled, “Efficacy and safety of a booster vaccination with 2 inactivated SARS-CoV-2 vaccines on symptomatic COVID-19 infection in adults: results of a double-blind, randomized, placebo-controlled, phase 3 trial in Abu Dhabi”, demonstrates the protective efficacy and safety of 2 inactivated SARS-COV2 covid vaccines. The authors have recruited healthy volunteers to evaluate the safety and efficacy of booster vaccinations that included the WIV04 and HBO2 and placebo group receiving only adjuvant. The manuscript is well written and is a significant contribution to the study. I have the following comments that will improve the presentation of the study and results

Comments:

-What is the difference between WIV04 and HBOB? What is the rationale behind using only 0.4 for one vaccine and 0.5 ug for another?

-why was different concentration of alum used in two different vaccines?

-How was the concentration of virion measured and how accurate the method is?

-Do you had patients in the placebo group who caught covid 19 and were asymptomatic?

-The authors should add graphs in the form of bar diagram/tables to make the manuscript easy to understand. Some of the tables from supplementary files can be moved to main manuscript.

Author Response

Kaabi et al of the manuscript titled, “Efficacy and safety of a booster vaccination with 2 inactivated SARS-CoV-2 vaccines on symptomatic COVID-19 infection in adults: results of a double-blind, randomized, placebo-controlled, phase 3 trial in Abu Dhabi”, demonstrates the protective efficacy and safety of 2 inactivated SARS-COV2 covid vaccines. The authors have recruited healthy volunteers to evaluate the safety and efficacy of booster vaccinations that included the WIV04 and HBO2 and placebo group receiving only adjuvant. The manuscript is well written and is a significant contribution to the study. I have the following comments that will improve the presentation of the study and results

Our response: We thank the reviewer for the summary of our study, as well as the positive comments.

Comments:

-What is the difference between WIV04 and HBOB? What is the rationale behind using only 0.4 for one vaccine and 0.5 ug for another?

Our response: We thank the reviewer for his/her questions. Based on various viral strains, the two businesses independently created the two vaccines. Because antigen detection reagents and the standards vary, so do virus concentrations.

-why was different concentration of alum used in two different vaccines?

Our response: We thank the reviewer for his/her question. Different adjuvant producers were used because the two vaccines were created independently by two different businesses. As a result, adjuvant concentration is not the same.

-How was the concentration of virion measured and how accurate the method is?

Our response: The content of virus S protein antigen in the vaccine was detected by double-antibody sandwich ELISA. In this detection method, anti-spike protein (S protein) antibody with neutralizing activity was selected as the coated antibody, the serum against the SARS-CoV-2 was selected as the detection antibody, and then HRP labeled IgG was used for color reaction. When testing for antigen content, both the sample and the antigen standard are tested, and the antigen content of each batch of sample is calculated based on the antigen content of the antigen standard. The method was methodologically validated in terms of accuracy, specificity, precision - repeatability, precision - intermediate precision, linear range and persistence. The antigen concentration of the two vaccines was determined by the above method, but the detection reagents, antibodies and standards were different.

We wholeheartedly concur with the reviewers' inquiries and think that this information would clarify our vaccines for readers. However, we didn't change the manuscript to include this method since we think that these details might be more appropriate to supplement in the previous clinical research articles.

-Do you had patients in the placebo group who caught covid 19 and were asymptomatic?

Our response: In fact, because of the large number of participants and limited conditions, it is not feasible to study asymptomatic infected people. This study did not focus on asymptomatic infections, which is also our great regret.

-The authors should add graphs in the form of bar diagram/tables to make the manuscript easy to understand. Some of the tables from supplementary files can be moved to main manuscript.

Our response: To make it easier for readers to grasp our findings, we combine reviewers' viewpoints and offer our findings in a variety of ways. Since clinical trial publications frequently contain a lot of data, we think it is more complete to offer the data as tables so that different readers may easily access the information they require. The original eTable5 was inserted inside the paper because we thought the conclusions it drew were more crucial.